# Zoonotic Implications of *Onchocerca* Species on Human Health

**DOI:** 10.3390/pathogens9090761

**Published:** 2020-09-17

**Authors:** Maria Cambra-Pellejà, Javier Gandasegui, Rafael Balaña-Fouce, José Muñoz, María Martínez-Valladares

**Affiliations:** 1Instituto de Ganadería de Montaña (CSIC-Universidad de León), 24346 León, Spain; mcamp@unileon.es; 2Departamento de Sanidad Animal, Facultad de Veterinaria, Universidad de León, Campus de Vegazana, 24071 León, Spain; 3Instituto de Salud Global de Barcelona (ISGlobal), 08036 Barcelona, Spain; javiergandasegui@gmail.com (J.G.); jose.munoz@isglobal.org (J.M.); 4Departmento de Ciencias Biomédicas, Facultad de Veterinaria, Universidad de León, 24071 León, Spain; rbalf@unileon.es

**Keywords:** *Onchocerca*, zoonosis, animal species, clinical signs

## Abstract

The genus *Onchocerca* includes several species associated with ungulates as hosts, although some have been identified in canids, felids, and humans. *Onchocerca* species have a wide geographical distribution, and the disease they produce, onchocerciasis, is generally seen in adult individuals because of its large prepatency period. In recent years, *Onchocerca* species infecting animals have been found as subcutaneous nodules or invading the ocular tissues of humans; the species involved are *O. lupi*, *O. dewittei japonica*, *O. jakutensis*, *O. gutturosa*, and *O. cervicalis*. These findings generally involve immature adult female worms, with no evidence of being fertile. However, a few cases with fertile *O. lupi*, *O. dewittei japonica*, and *O. jakutensis* worms have been identified recently in humans. These are relevant because they indicate that the parasite’s life cycle was completed in the new host—humans. In this work, we discuss the establishment of zoonotic *Onchocerca* infections in humans, and the possibility of these infections to produce symptoms similar to human onchocerciasis, such as dermatitis, ocular damage, and epilepsy. Zoonotic onchocerciasis is thought to be an emerging human parasitic disease, with the need to take measures such as One Health Strategies, in order to identify and control new cases in humans.

## 1. Introduction

The genus *Onchocerca* comprises several parasitic nematode species [1], which are transmitted by arthropod vectors, namely, blackflies belonging to the genera *Simulium* and *Culicoides* [2], and it has been suggested that the biting habits of these blackflies influence the presence and distribution of microfilariae in the subcutaneous tissues of the hosts [3]. Until now, several *Onchocerca* species have been identified, and ungulates seem to be the main hosts, although the parasite is also found in canids, felids, and humans [4]. Each larva stays as a single male or female and grows until it becomes a mature worm, which remains separated in the skin or collected together in subcutaneous fibrous nodules because of the immunological reaction of the host; females produce microfilariae, which invade the skin, travel in it, and very often reach the eye [1,5]. As a result, onchocerciasis usually causes dermatitis and ocular damages, but epilepsy has also been reported in humans [6]. In recent years, *Onchocerca* species infecting animals were found in human tissues. In this work, we review the *Onchocerca* infections in animals and their potential zoonotic character, as well as analyzing whether the filarial worms belonging to the genus *Onchocerca* are able to produce clinical signs in humans, rendering onchocercasis an emerging zoonotic disease.

## 2. How Many Definitive Hosts Are Parasitized by *Onchocerca spp.*?

The genus *Onchocerca* comprises more than 30 species which mainly infect ungulates and have a worldwide distribution [4]. The only known host exceptions are carnivores, canids, felids, and humans [4]. In general, onchocerciasis is more prevalent in adult rather than juvenile individuals, because of the large prepatency period of the infection [5]. *Onchocerca* infections may have been underestimated, particularly in wildlife, where infected animals are easier to be preyed upon, especially when the *Onchocerca* nodules are found in the joints, making movement difficult [7]. However, *Onchocerca* control in wildlife is difficult because of the ubiquity of the adult insect vectors [8], the need of larviciding of vector breeding sites [6], and the migratory patterns of some hosts, leading to the expansion of *Onchocerca* species [9]. The most common *Onchocerca* species identified so far, related to their main vectors, hosts, and geographical distribution, are described in Table 1.

Some authors emphasize that knowing the evolutionary history of a parasite and its spread among its hosts is crucial to control the disease it causes [104]. The origin of the genus *Onchocerca* and its principal evolution may have been started in Africa. This hypothesis is based on the fact that the most primitive species were found in Africa, the continent with the highest diversity of *Onchocerca* species [105]. From these *Onchocerca* primitive African species, diversification may have occurred until reaching the current species. There are two main hypotheses about the origin of these species; the first one postulates that they evolved via co-speciation with the host, and the second one indicates that they could have been captured via horizontal transfer from other host species, naming the latter host-switching. As the host specificity in some *Onchocerca* species is not as strong as was thought to be [53], it seems feasible that multiple host-switching events between hosts have historically occurred [106]. Some authors support the idea that the domestication of animals may have had an important role in those horizontal transfers leading to different *Onchocerca* species, parasitizing both animals and humans [12] (Figure 1). Thus, onchocerciasis could be considered a zoonosis with different potential hosts [104].

*Onchocerca* species identification has traditionally been done through morphological examinations [39]. The number, height, and longitude between the crests in the outer cuticle, the shape of the anterior tip, the body thickness, and the presence or absence and the shape of the striae from the inner cuticle are used to achieve the morphological distinction between *Onchocerca* species [39]. Difficulties during worm extraction could complicate the later morphological identification; therefore, authors such as Verocai et al. [7] support the combination of classical and molecular tools to characterize the specimens collected. Regarding the molecular methods, the analysis of mitochondrial cytochrome c oxidase subunit 1 (COI) and 12S ribosomal RNA gene sequences are the most used for species identification [39]. 

In wild ruminants such as reed deer, moose, and sika deer, among others, *Onchocerca* has been described traditionally in Central and Eastern Europe, but recent evidence shows that they have a broader geographic distribution [8]. Some studies indicate that *Onchocerca* species located in insular regions such as Japan showed more primitive morphological characters than species from the continent, as a result of their geographical isolation [94]. In wild ruminants, adult worms have generally been identified in subcutaneous nodules located in the dorsal region and flanks [8], along the metatarsus and metacarpus [107], in the posterior dorsal region and thighs [58], and in the upper parts of the extremities [108]. However, in some cases, adult parasites were found in these hosts without being included within a nodule [47,94]. Regarding the microfilariae, they were located inside the hind limbs [8], the ventral zone [58], ears and muzzles [109] of the hosts. 

*Onchocerca* infections affecting domestic animals such as cattle have usually been identified in slaughterhouses [13]. *Onchocerca armillata* has a specific location in the host, very different from other species, in the tunica intima of the aorta [10]. On the other hand, the remaining species affecting cattle such as *O. gutturosa* [53], *O. ochengi*, and *O. dukei* [15], among others, generally parasitized the cervical ligaments [12] and the connective tissues of the ventral and thoracic areas of the cattle [10,43,53].

In horses, the appearance of skin nodules is often asymptomatic, and it is not usually noticed by owners, but it could also manifest with dermatitis, impaired function of the ligaments, and even blindness [19]. Adult *Onchocerca* parasites are usually located coiled in the ligamentum nuchae of horses [30] and within nodules infecting the connective tissue of the flexor tendons or the suspensory ligament of the fetlock [110]. Several *Onchocerca* species also have been found in the locations described above, but infecting camels and dromedaries [12]. Until now, what is known is that *O. fasciata* is a species specific in camels [50], being the most prevalent species in this host [48]. Camelidae are valuable animals in some areas of the world, as these animals are used as transport and their meat is used for human consumption; therefore, onchocerciasis have a detrimental impact associated with a loss of commercial value [48] and major issues in public health [54]. 

*Onchocerca lupi* was described for the first time in the ocular tissues of wolves [73], and since then, it has mainly been reported in dogs [74,111]. In the latter, *O. lupi* adult worms are generally found in the ocular nodules, which could result in eye damage, such as conjunctivitis, swelling, exophthalmia, and vision loss [74]. Microfilariae seem to aggregate in specific body areas in the head, specifically ears and nose, or in the inter-scapular region [111]. However, *O. lupi* was also found infecting cats and again showing an ocular tropism [75,112]. 

In humans, onchocerciasis is caused by *O. volvulus* [113]. The adult worms are distributed in subcutaneous nodules under the skin, while microfilariae are generally found on the hips, shoulders, and the lower parts of the body [113]. Some people do not present any clinical signs while others exhibit itchy skin rashes and vision disorders [113]. Recently, several authors also described a specific type of epilepsy as a disorder related to onchocerciasis [114,115,116]. Human onchocerciasis is more prevalent in fertile regions of developing countries such as Venezuela and Brazil [113], Tanzania [114], Uganda [115], or South Sudan [116], causing a high morbidity burden [104]. 

Female blackflies of the genus *Simulium* are the best-known vectors that transmit *Onchocerca* microfilariae [39]. *Simulium* larvae inhabit areas with running freshwater or rivers, where they may develop to the following stages [39]. The identification of infective larvae in wild-caught blackflies is important in order to assess onchocerciasis transmission rate and putative natural *Onchocerca* vectors [38]. The screening of transmission vectors could be done by the morphological identification of microfilariae or via molecular methods, detecting their DNA by PCR amplification within vectors [48]. Although many species within the genus *Simulium* and *Culicoides* are likely to be putative vectors of *Onchocerca spp.* (Table 1), many of them may be still unidentified. The wide geographical distribution of *Onchocerca* vectors indicates favorable environmental conditions for spreading infection [5].

## 3. Does Onchocerciasis Have a Zoonotic Character?

The incidence of zoonotic diseases increases when humans live in close contact with animals, such as when both hosts overlap in a geographic region [117]. Thus, it is important to consider humans together with animals and the environment as a whole, which is emphasized in the One Health Strategy, in order to prevent and control the emergence of zoonotic diseases. Emerging zoonotic diseases impact on public health and socioeconomic aspects of the global population [117]. 

So far, 40 cases of *Onchocerca* species infecting animals have been described in humans worldwide, and all of them are within the Holarctic region [68,96,118,119,120,121,122,123,124,125,126] (Figure 2). Among these cases in humans, *O. lupi* was the most prevalent species identified, followed by *O. dewittei japonica* and, to lesser extent, *O. jakutensis*, *O. gutturosa*, and *O. cervicalis*. Moreover, there are several cases where *Onchocerca* parasites infecting animals were found in humans, but the species was not identified [127,128,129,130,131,132,133] or was only suspected without confirmation [134,135,136].

The first human case unequivocally caused by *O. lupi* was found infecting the subconjunctival region of a person in Turkey [76]. Since then, additional findings of this species were also reported in the ocular tissues of people from Turkey, Tunisia, and Iran [80,81,82]. The first report of *O. lupi* in the United States was published in 2013, and the worm was located in the individual’s upper cervical spinal cord [137]; two years later, another report involving the same tissues was identified in the same country [119]. Most findings of *O. lupi* in humans described between 1965 and 2014 were reviewed by Grácio et al. [120]. However, a few of them regarding other species were not included in that review, for example, *O. dewittei japonica*, which was isolated in individuals from Japan from different body parts such as hand [41], infraclavicular region [138], and knee [139]. Other examples not reported previously in the review were the case of a woman in Australia with multiple onchocercal nodules on the neck and face [140] or the case of another woman in the United States with a subdeltoid mass caused by *O. gutturosa* [66].

Since 2015, 12 reports of *Onchocerca* species infecting animals have been described in humans [68,96,118,119,121,122,123,124,125,126]. Five of them were caused by *O. lupi*, which was found in the upper cervical spinal cord, conjunctiva, head, and forearm of five individuals diagnosed in the United States [121,122]. Six more cases were reported in Japan, caused by *O. dewittei japonica*, which was isolated from the upper extremities in five individuals [123,124,125,126] or from the head of one person [96]. The most recent case was caused by *O. jakutensis* and was found in the vitreous body of the eye of a man from Poland [68]. These cases were generally caused by immature adult female worms, most of them with no evidence of fertile adult worms, and thus no microfilariae seemed to be produced [39]. However, to determine the fertile status of these worms, it was important to remove them intact from the nodules or tissues where they were located [68]. Sometimes, the worms were damaged during their extraction, and therefore, the chances of obtaining a good quality sample for morphological identification were reduced [68]. In this case, molecular tools such as PCR can be used for species identification [7].

Koehsler et al. [140] described a case caused by *O. jakutensis* in an Austrian woman with lupus erythematosus and with multiple nodules on the neck and face. Although the presence of fertile worms was not shown in those nodules, it is thought that the autoimmune disease could have favored the appearance of multiple nodules developed over several years, showing that worm reproduction was possible, and this was the first human case involving multiple nodules. However, in a few reports, the presence of fertile adult worms from *Onchocerca* species infecting animals was confirmed in human tissues. Dudley et al. [119] identified a mass causing cervical spinal cord compression in a girl from the United States, containing a gravid *O. lupi* female with uteri filled with microfilariae. Uni et al. [96] described an infection caused by a male *O. dewittei japonica* and located in the head of an individual in Japan; spermatozoids were identified in the seminal vesicle, showing the maturation of the male *O. dewittei japonica* in humans [96]. In addition, Bergua et al. [118] reported an eye infection caused by *O. lupi*; in this case, the worm was removed by surgery, and one month later a new nodule appeared again in the patient’s face, confirming the presence of *O. lupi*. These cases are relevant because they show that the parasite’s life cycle was able to be completed and therefore the establishment of zoonotic *Onchocerca* infections is possible in humans. 

Considering the cases previously mentioned, *O. lupi*, *O. dewittei japonica*, and *O. jakutensis* are proposed as zoonotic species that could reproduce and complete the parasite’s life cycle within the human body. This phenomenon was previously suggested by Uni et al. [96], postulating that accidental zoonotic *Onchocerca* species could switch from one host to another through their vectors, adapting to humans and becoming a new human parasitic disease. Moreover, Grácio et al. [120] and Tahir et al. [141] proposed in their reviews that *O. lupi* cases in humans should be contemplated as a current emerging zoonotic disease.

## 4. Which Factors Are Involved in the Transmission of Animal *Onchocerca* Species to Humans?

There are several factors that favor the transmission of *Onchocerca* species infecting animals to humans. The high prevalence of the causative agents in the host animals is one of those factors [138]. A prevalence of 92% of *O. dewittei japonica* in wild boars in the Oita region of Japan was associated with several cases in humans in the same area [138]. The expansion of wild animal habitats due to alterations in climate, deforestation, and urbanization appears to be an important factor that favors contact between animals and humans [96]. On the other hand, climate determines the geographical distribution of potential *Onchocerca* vectors, and changes in the environment, such as accelerating climate warming, may modify the ranges where these vectors can live [142]. Modifications in any of these factors could lead to changes in *Onchocerca* species distribution [53], and as a consequence, altered patterns of exposure could lead to contact with new hosts, animals or humans [7].

Since the most frequent *Onchocerca* species infecting animals found in humans are *O. dewittey japonica* from Japanese wild boar, *O. lupi* from carnivores (dogs or wolfs), and *O. jakutensis* from red deer [68], it is important to evaluate the onchocerciasis infection rate in these hosts, especially if they share the same habitat with humans. In order to define areas of risk effectively, it is essential to plan public health strategies to prevent the transmission of *Onchocerca* species infecting animals to humans [126]. Although it is not clear that either an increasing number of zoonotic reports or a better diagnostic is being performed, or both [68], zoonotic onchocerciasis is likely to occur more widely than expected.

## 5. What Are the Lesions and Clinical Signs that *Onchocerca* Species Infecting Animals Cause in Humans?

Zoonotic onchocerciasis is characterized by lesions observed as subcutaneous nodules developed around adult worms in various parts of the body and, in some cases, reaching the ocular tissue [39]. The common clinical sign related with this infection is local swelling, so the pathology of the infection does not depend on *Onchocerca* species but on the invaded tissue [118]. As previously described, most of the *Onchocerca* species infecting animals but found in humans are detected by the appearance of subcutaneous nodules [124]. Nodules can grow in size, swell, and itch, and some of them are painful [124]. So far, the most effective treatment is the surgical removal of the worms [68]. Then, species identification is performed according to the cuticular morphology of the adult worms or by PCR [143]. 

However, three *Onchocerca* species infecting animals were able to reach the ocular tissues. *O. lupi* was found within the conjunctiva [76,80,81,82], the superior rectus muscle of the eye [122], and the anterior chamber of the eye [118,137], while *O. jakutensis* was located in the vitreous body of the eye [68] and *O. cervicalis* in the cornea [26]. Although the most characteristic lesion is the presence of an ocular mass, other cases may be accompanied by milder signs such as blurred vision, redness, irritation, or itchiness, or more severe signs such as swelling, eyelid drooping, conjunctival hyperemia, and pain [76,80,118,122,137]. Some authors have considered the possibility that the ocular invasion of *O. lupi* in humans could be underestimated, being more widespread than believed, as some cases could have been misdiagnosed as other species [80]. *O. lupi* is also able to invade the Central Nervous System (CNS) of its hosts, leading to lesions in the upper cervical spinal cord or neck pain and stiffness, but also, less frequently, sore throat, dysphagia, and fever [119,121,122,137].

### 5.1. Is It Possible That Onchocerca Species Infecting Animals Produce Clinical Symptoms Such as Blindness in Humans?

Onchocerciasis caused by *O. volvulus* is the world’s second leading cause of infectious blindness [144]. However, it is generally accepted that the true prevalence of blindness associated to onchocerciasis is underestimated [144]. Previous studies have suggested that onchocercal blindness is more common in savanna regions of Africa compared with forest areas, where dermatological symptoms are predominant [145,146], and some studies are being conducted to elucidate the parasite differences between those areas. The microfilariae produced by female *O. volvulus* migrate to the ocular tissues in the human host and invade every part of the eye except the lens. Living microfilariae localized in the eye do not cause inflammation [147], the ocular damage appears as a result of the host immune response to antigen release from degenerating and dead microfilariae [148]. The risk of developing blindness increases with greater exposure to a high microfilariae load [148]. The results are impaired vision from corneal opacities, cataract, chorioretinal degeneration, optic atrophy, and in severe cases permanent blindness with subsequent chronic disability and reduced life expectancy [147,149]. 

In animals, it is known that adult worms of *O. lupi* could cause vision loss in dogs [74] and *O. cervicalis* in horses [30]. Adult worms of *O. lupi*, *O. cervicalis*, and *O. jakutensis* were found invading human ocular tissues, but blindness was not diagnosed in any individual [26,68,118]. Thus, it seems that the *Onchocerca* species infecting animals but found in the human ocular tissues involved adult parasites present in the conjunctiva and the cornea, as it happens in their natural hosts [30,74]. This contrasts with *O. volvulus* infection in humans where microfilariae are found in the ocular tissues and are responsible for the ocular damage [148,149]. 

### 5.2. Is It Possible That Onchocerca Species Infecting Animals Produce Clinical Disorders Such as Epilepsy in Humans? 

It has been estimated that onchocerciasis-associated epilepsy (OAE) affected between 300 and 400,000 people worldwide in 2015 [150]. The onset of OAE is between the ages of 3 and 18 years old, and OAE is an important cause of mortality among children and adolescents in onchocerciasis endemic areas with ongoing transmission [151]. Recent epidemiological studies strongly suggest that the “parasite is able to directly or indirectly trigger epilepsy” [152,153,154,155]. A post mortem study done in nine children who died of OAE including nodding syndrome showed signs of neuroinflammation and Tau deposits but no signs of parasitic infection [156].

The mechanism through which *O. volvulus* causes epilepsy remains unknown [6]. One of the most important challenges to study OAE etiology is the need to perform autopsies for subsequent histopathological analysis, which can be extremely difficult in most rural African regions. Different hypotheses about OAE etiology have been postulated: direct infection of the CNS by microfilariae [157], autoimmune reactions resulting in neurotoxicity [158], and the transmission of a neurotropic virus together with the *Onchocerca* microfilariae [159]. These hypotheses were published in the American Journal of Ophthalmology [157], Science Translational Medicine [158], and Annals of Tropical Medicine and Parasitology [159] journals, respectively. However, most of these hypotheses suggest that there is a factor related to onchocerciasis that triggers the episode of epilepsy. 

The first theories published on the etiology of OAE suggested that direct infection of the CNS by *O. volvulus* microfilariae could be the cause of OAE onset [157,160]. However, microfilariae nor *O. volvulus* DNA were observed in recent studies in cerebrospinal fluid (CSF) nor brain of persons with OAE [161].

The hypothesis relating OAE and immunological factors of the host is gaining importance. Considering that these factors are known to be involved in the development of blindness, they may also be associated with the onset of epilepsy [158]. Therefore, epilepsy could be the result of an autoimmune reaction of antibodies against *O. volvulus*, which results in a neurotoxic reaction [162]. The entrance of *O. volvulus* microfilariae in the human host triggers the activation of the immune system, increasing the levels of several cytokines that migrate from the periphery into the CNS, promoting neuroinflammation [160]. As a consequence, antibodies against *O. volvulus* proteins could cross the blood–brain barrier and recognize antigens from the CNS of the host. This may trigger a neurotoxic reaction, which could promote the onset of OAE [160]. Several proteins found in the CNS have been proposed as cross-reacting antigen candidates, such as voltage-gated potassium channels (VGKC) [163], leiomiodin-1 (LM-1), and the human protein deglycase DJ-1 [158]. Further studies are needed to elucidate the underlying mechanism of immunological mediated toxicity.

Finally, Colebunders et al. [164] proposed a new hypothesis in which an unknown neurotropic virus or an endosymbiont of the parasite transmitted together with *Onchocerca spp.* and through the same vectors could cause OAE. This hypothesis was based on the findings by Mellor and Boorman [159], who observed that the vector *Culicoides nubeculosus* transmitted the bluetongue virus in ruminants only when *O. cervicalis* microfilariae where found together in the vector. In this sense, *Wolbachia* is known to be an endosymbiont bacterium for some *Onchocerca* species, including *O. volvulus* [148,165]. *Wolbachia* antigens can contribute to the appearance of some of the clinical signs related to onchocerciasis such as ocular damage [148,165], but it is unknown if they could also contribute to OAE. 

Regarding *Onchocerca* species infecting animals, Cantey et al. [122] reported the presence of *O. lupi* in the cervical spinal canal of humans, evidencing that some *Onchocerca* species infecting animals could reach human CNS, but no recent study has found the parasite in the CSF. However, until now, epileptic seizures have not been reported in zoonotic onchocerciasis. Some zoonotic reports, with fertile female worms involved, suggested that this could be a possibility if the biological life cycle is completed and microfilariae are produced [96,118,119]. Therefore, some microfilariae could reach the CSF or trigger a neurotoxic reaction in the host, leading to epileptic seizure. 

## 6. Concluding Remarks

*Onchocerca* species infecting animals have been found invading human tissues, with *O. lupi*, *O. dewittei japonica*, and *O. jakutensis* being the most common species. The high prevalence of causative agents in wildlife and domestic animals and the extension of their habitats due to alterations in climate and urbanization seemed to be crucial factors to bring *Onchocerca* parasites infecting animals closer to humans [96], increasing human-parasite contact, and thus leading to zoonotic transmission. Most of these invasions seem to produce only immature adult worms [39], but several cases with fertile worms involved have been recently reported. Those cases involved the three species mentioned above, *O. lupi*, *O. dewittei japonica*, and *O. jakutensis*, and we postulate them as candidate species that could reproduce and maintain the parasite’s life cycle within the human body. As this has occurred before, it is possible that *Onchocerca* species infecting animals could switch from their primarily hosts to humans in the future, causing symptoms similar to human onchocerciasis, such as dermatitis, eye damage, and even epilepsy. All of this makes zoonotic onchocerciasis an emerging human parasitic disease, with the need to take measures, such as One Health Strategies, in order to identify and control new cases in humans. 

## Figures and Tables

**Figure 1 pathogens-09-00761-f001:**
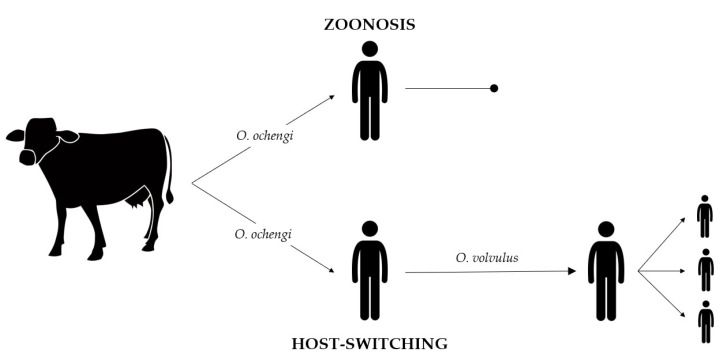
Host-switching event leading to the appearance of new *Onchocerca* species. *Onchocerca* microfilariae from cattle, such as *O. ochengi*, are transmitted through blackflies to a new host, humans. In these new hosts, *O. ochengi* remain without being transmitted to other humans, leading to transmission interruption (rounded arrow), called zoonosis. Another possibility is that *O. ochengi* may have settled in humans and evolved to new species, such as *O. volvulus*. In that case, *O. volvulus* is transmitted through blackflies between humans (pointed arrow). This event is called host-switching.

**Figure 2 pathogens-09-00761-f002:**
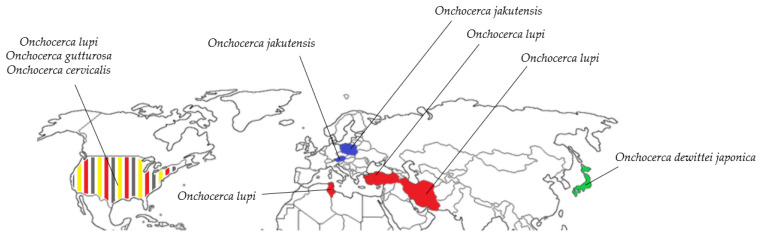
Geographical distribution of *Onchocerca* species infecting animals identified in humans through the Holarctic region. The species found are the following: *O. lupi* (red), *O. dewittei japonica* (green), *O. jakutensis* (blue), *O. gutturosa* (grey), and *O. cervicalis* (yellow).

**Table 1 pathogens-09-00761-t001:** Description of the most common *Onchocerca* species identified parasitizing animals and humans, related to their main vectors, hosts, and geographical distribution.

*Onchocerca* Species	Vectors	Hosts	Geographical Distribution
*O. armillata*	Unknown [10]	Cattle [11], buffaloes [11], dromedaries [12], goats [11]	Ghana [13], Iran [14], Senegal [15], Sudan [12], Sierra Leona [16], Tanzania [17], Nigeria [3], Cameroon [10], India [18]
*O. boehmi*	Unknown [19]	Horses [19]	Austria [20], Iran [21], Italy [19]
*O. cervicalis*	*Culicoides nubeculosus* [22], *C. variipennis* [23]	Horses [24], ponies [25], humans [26], donkeys [27]	Australia [24], Japan [28], United Kingdom [22], Holland [29], Canada [30], United States [31], Brazil [32], Poland [33], Spain [33], Egypt [27]
*O. cervipedis*	*Simulium venustum* [34]	White-tailed deer [35], black-tailed deer [35], moose [9], caribou [9]	Canada [35], Costa Rica [36], United States [9], Alaska [9]
*O. dewittei dewittei*	Unknown	Wild boar [37]	Malaysia [37]
*O. dewittei japonica*	*Simulium bidentatum* [38], *S. arakawae*, *S. japonicum*, *S. oitanum*, *S. quinquestriatum*, *S. rufibasis* [39]	Wild boar [40], humans [41]	Japan [40]
*O. dukei*	*Simulium hargreavesi*, *S. vorax*, *S. damnosum s.l.* [42]	Cattle [43]	Zambia [44], Togo [45], Cameroon [43]
*O. eberhardi*	*Simulium arakawae*, *S. oitanum*, *S. bidentatum* [46]	Sika deer [47]	Japan [47]
*O. fasciata*	*Culicoides puncticollis* [48]	Dromedaries [49], camels [50]	Sudan [51], Somalia [51], Jordan [52], Saudi Arabia [53], Iran [54], China [55], Mongolia [48]
*O. flexuosa*	*Prosimulium nigripes*, *Simulium ornatum* [56]	Antelope [57], reed deer [58], Roe deer [59]	Uganda [60], Germany [61], Spain [8], Sweden [5], Slovakia [62]
*O. gutturosa*	*Culicoides spp.*, *C. kingi* [63]	Cattle [43], camels [64], dromedaries [12], horses [65], humans [66]	Iran [14], Senegal [15], Australia [64], Togo [45], Sudan [12], Sierra Leona [16], Cameroon [43], Turkey [67], India [18]
*O. jakutensis*	Unknown	Red deer [58], humans [68]	Germany [53], Italy [69], Austria [70], Switzerland [71], Poland [68]
*O. lupi*	*Simulium sp.* [72], *S. tribulatum* [7]	Wolf [73], Dogs [74], Cats [75], Humans [76]	Georgia [73], Germany [77], Greece [78], Hungary [79], Tunis [80], Turkey [81], Iran [82], Portugal [75], United States [83], Spain [84]
*O. ochengi*	*S. damnosum s.l.* [85]	Cattle [43]	Burkina Faso [86], Senegal [15], Mali [87], Sierra Leona [16], Togo [45], Cameroon [43], Ghana [88]
*O. ramachandrini*	*Simulium damnosum s.l.* [89]	Warthogs [90]	Cameroon [90], Uganda [89]
*O. reticulata*	*Culicoides nubeculosus Meig* [91]	Horses [92], donkeys [27]	France [93], Australia [92], United States [65], Egypt [27]
*O. skrjabini*	*Simulium arakawae*, *S. oitanum*, *S. bidentatum* [46], *S. japonicum*, *Prosimulium sp.* [28]	Japanese serow [94], sika deer [47]	Japan [47,94]
*Onchocerca spp. type I*	*Simulium bidentatum* [38], *S. sigrogilvum* [95]	Wild boar [38], cattle [95]	Japan [38], Thailand [95]
*O. suzukii*	*Simulium japonicum*, *Prosimulium sp.* [28]	Japanese serow [94]	Japan [94]
*O. takaokai*	*Simulium bidentatum* [96]	Wild boar [96]	Japan [96]
*O. volvulus*	*Simulium sp.* [97]	Humans [97]	Brazil [98], Guatemala [99], Uganda [100], Tanzania [101], Democratic Republic of the Congo [102], Yemen [103]

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
