# Peer review of "Zoonotic Implications of Onchocerca Species on Human Health"

_pathogens, 2020, doi:10.3390/pathogens9090761_

Round 1

Reviewer 1 Report

In this manuscript the authors review the rather few known cases of zoonotic Onchocerciasis, involving species of Onchocerca other than O. volvulus. The latter is believed to have no animal reservoir and is the major cause of human Onchocerciasis, one of the recognized neglected tropical diseases. The authors discuss the medical implications and relevance of these cases. The subject is interesting and the manuscript is timely. The authors compiled and worked through an impressive collection of references, which is most useful. Unfortunately, the writing is a bit uninspiring, solely descriptive and not sufficiently comparative. I think that the manuscript in its current form sells the extensive work the authors did much worse than it could. I have the impression that sections 1 and 2 were written by a different person than sections 3 to 5. The connections between the two parts could be improved and there is unnecessary repetition between them. Mainly sections 1 and 2 need language improvement. Some examples of confusing statements are given under specific points. Although the necessary information is provided in the current manuscript, I think by restructuring and a more precise language the points made by the authors could be made more clear. For example, it does not always become clear which developmental stages are responsible for the pathology of the different Onchocerca species, in particular blindness, in the natural hosts. While this is clearly stated for e.g. O volvulus (microfilariae) and O. lupi (the adult worms located in the eyes) it does not become clear for other species and hosts, e.g. horses (lines 103-105). Also, I would highly welcome a more specific discussion of the question in which ways the zoonotic Onchocerca infections in humans resemble O. volvulus infections and in which ways they resemble infections of the corresponding Onchocerca species in its natural host (considering e.g. location of the nodules and the mf and the possible mechanism of pathology). This information is interesting not only for treatment but also for basic science because it might give hints about what aspects of an Onchocerca infection are controlled by the host and the parasite, respectively.

Specific points:

1) Throughout the manuscript: italicize all genus and species names (e. g. lines 54, 90, from line 134 on, no genus or species names are italicized).

2) Lines 32-34: This sentence is not clear. Do the authors talk about the place in the body the black flies tend to attack and the mf are mostly found, the time of the day or the year the black flies predominantly bite, or both?

3) Line 52 and heading to table 1: What are "most representative Onchocerca species"? Do the authors mean the most commonly found species?

4) Table 1: Maybe it is worth adding Onchocerca sp. type I since it was found in black fly species known to bite humans in multiple countries (see Reference 39 and Saeung et al. (2020) Natural infections with larvae of Onchocerca species type I in the human-biting black fly, Simulium nigrogilvum (Diptera: Simuliidae), in western Thailand. Acta Tropica 204, 105344 [I am not an author of this publication]).

5) Lines 60 and 91: what do you mean with "primitive"?

6) Line 125: This sentence is unclear. Simulium are not "vectors of blood sucking insects" they are blood sucking insects and they are vectors for parasites.

Reviewer 2 Report

  1. Introduction:

This manuscript was very interesting, please find details on revisions below. Please pay attention to citation, italics issues and the phrasing around animal Onchocerca spp.

Line 38: should read…causes…

Lines 40: Onchocerciasis is the pathogenic condition caused by several species of the filarial

            nematodes or worm within the genus, Onchocerca found invading human tissues.

            Revise sentence to reflect this statement.

Line 41: perhaps change sentence: …whether the filarial worms belonging to the genus,

            Onchocerca are able to…. Ok, after getting through most of the paper, I think I

            Understand why the authors are using the word animal, they are not referring to the

            filarial worms, they are referring to those taxa that have been characterized as strictly

            zoonotic infections. The authors need to go through the entire paper and clarify this

            point wherever they use the phrase animal Onchocera spp. Perhaps change to

            zoonotic (sensu lattu) Onchocerca spp.

  1. How many definitive hosts are parasitilzed by Onchocerca spp.?

Line 45: citation after distribution.

Line 46: should read: …more prevalent in adult rather than juvenile individuals…

Line 47: Whose prevalence? Clarify

Line 50: by control, the authors are referring to adult fly control, however, most blackfly

            control is on the larval stage of the black fly, authors should clarify control here.

* very nice job summarizing vectors, hosts and geographical distributions in Table 1.

Line 57: citation needed after produces or combine with next sentence, e.g., …the disease it

            causes (not produces) and argue that….

Line 61: The sentence beginning with Then…is confusing, clarify, a bit vague.

Line 80: citation needed after examinations.

Line 82: citation needed after species.

Line 88: Perhaps change to: In wild ruminants, such as reed deer, ….

Line 98: Perhaps change to: Regarding domestic animals such as cattle, …. And cite this

            sentence.

Line 99: Spell out Onchocerca to lead off the sentence.

Line 100: maybe give a few examples of the other species that affect cattle

Line 113: see line 99

Line 114: who reported it in dogs, cite this source.

Line 119: citation needed after volvulus.

Line 121: citation needed after body. Perhaps revise next sentence: Some people do not

present with any clinical symptoms, while others exhibit…

Line 124: authors should clarify where or what are the fertile regions, e.g., West Africa or

            specific countries and fertile referencing agricultural regions?

Line 125: citation needed after parasites.

Line 126: begin sentence, Simuliium larvae inhabit…and revise end of sentence.

  1. Does onchocerciasis have a zoonotic character?

Line 136: citation needed after region.

Lines 137-138: revise this sentence, it is not well written, perhaps split into two sentences.

Line 141: citation needed after region. Also, next sentence, to what are the authors referring

            by saying This contrasts…? What opposite area? This sentence is confusing.

Line 142: among which reports, there were no citations previously, refer to previous

            comment.

Lines 142-144: Italicize scientific names of Onchocerca species.

Figure 2. See comment above on italicizing scientific names.

Lines 151-316: the authors neglected to italicize any scientific names in the second half of the

            paper, please pay attention to details and italicize.

Line 155: where was this second report identified, which country, still the USA?

Line 162: Please correct this misuse of animal, see comments on lines 41 and cite,

            citation is missing.

Line 171: citation needed after located.

Line 173: citation needed after identification. Would this be citation #143?

Change number in next section to 4 and rephrase (see below).

  1. Which factors are involved in human infections of Onchocerca species?

Line 207: see line 41 comment

Change number in next section to 5 and rephrase.

  1. What are the lesions and clinical symptoms with Onchocerca spp infections in humans? See

line 41 comments.

Line 218: citation after tissue.

Line 219: See line 41 comments and cite.

Line 237: citation after blindness. What situation? Infectious? Clarify.

Line 251: see line 41 comments

Line 264: cite these hypotheses after epilepsy.

Line 266: cite where these theories were published after onset.

Line 267: any literature support for this statement?

Line 273: cite the source for this hypothesis

Lines 279-280: any lit support for this statement or conjecture?

Line 280: This what

Round 2

Reviewer 1 Report

I thank the authors for considering my comments in their revisions. The manuscript is now much more clear and understandable. The first and the last paragraph, the manuscript could still benefit from some language improvement. I made a few suggestions under "specific points". However, I am not a native English speaker either and I leave it to the authors to what extent they want to revise the wording. The paragraphs are understandable, such that changes are not essential. In some places spaces between words are missing and I found a few instances where "Onchocerca" was not italicized (see specific points).

Specific points:

1) Line 31: "genera" (plural) not "genus"

2) Line 38: "[1,5]. As a..." (space missing)

3) Line 41: "their" instead of "its"

4) Line 43: I suggest "rendering" instead of "becoming"

5) Line 61: "[106]. From..." (space missing)

6) Line 146: "identification was" (space missing between the two words).

7) Lines 148, 163, 180, 193: "italicize "Onchocerca".

8) Line 296: "[159]. As a..." (space missing)

9) Lines 321, 322: I suggest rewording: "Onchocerca species infecting animals have been found invading human tissues, with O. lupi, O. dewittei japonica and O jakutensis being the most common species."

10) Line 326: I suggest rewording: "Most of these invasions seem to produce only immature adult worms [39], but several...." (the adults to not produce invasions, L3 invade)

Reviewer 2 Report

Just fix one item: line 98 or somewhere around there, start sentence with spelling out Onchocera, do not start sentence with O. 

Round 3

Reviewer 1 Report

I am fully satisfied with the changes the authors made and now recommend to accept this manuscript for publication.